# Preeclampsia: The Relationship between Uterine Artery Blood Flow and Trophoblast Function

**DOI:** 10.3390/ijms20133263

**Published:** 2019-07-02

**Authors:** Anna Ridder, Veronica Giorgione, Asma Khalil, Basky Thilaganathan

**Affiliations:** 1Vascular Biology Research Centre, Molecular and Clinical Sciences Research Institute, St. George’s University of London, London SW17 0RE, UK; 2Fetal Medicine Unit, St. George’s University Hospitals NHS Foundation Trust, Blackshaw Road, London SW17 0RE, UK

**Keywords:** preeclampsia, uterine artery, maternal cardiovascular system

## Abstract

Maternal uterine artery blood flow is critical to maintaining the intrauterine environment, permitting normal placental function, and supporting fetal growth. It has long been believed that inadequate transformation of the maternal uterine vasculature is a consequence of primary defective trophoblast invasion and leads to the development of preeclampsia. That early pregnancy maternal uterine artery perfusion is strongly associated with placental cellular function and behaviour has always been interpreted in this context. Consistently observed changes in pre-conceptual maternal and uterine artery blood flow, abdominal pregnancy implantation, and late pregnancy have been challenging this concept, and suggest that abnormal placental perfusion may result in trophoblast impairment, rather than the other way round. This review focuses on evidence that maternal cardiovascular function plays a significant role in the pathophysiology of preeclampsia.

## 1. Introduction

Maternal uterine artery blood flow is one of the critical factors that contribute to the preservation of the intrauterine environment, which permits normal placental function to support fetal growth and development. This is so, not only because maternal blood carries nutrition and removes waste, but also because oxygen delivered to the developing fetoplacental unit is directly limited by uterine blood flow. Spiral arterioles that perfuse the intervillous space undergo significant morphologic changes during this process, with uterine vascular adaptations resulting in five to 10-fold dilatation to meet the requirements of the fetoplacental unit [1]. It has long been believed that inadequate development of the uterine vasculature may be a consequence of primary defective placentation, which may lead to the development of both preeclampsia and fetal growth restriction. Understanding the relationship between uterine artery blood flow and placental development is fundamental to understanding normal placentation and its disruption in both preeclampsia and fetal growth restriction. This review focuses on the relationship between uterine artery blood flow and the trophoblast function, and discusses the insights provided into the pathophysiology of preeclampsia.

## 2. Uterine Artery Blood Flow Assessment

Maternal uterine arteries can be readily and reliably identified via ultrasound by the use of a color Doppler and the pulsatility index (resistance to blood flow), assessed concurrently with a pulsed wave Doppler. Resistance to blood flow in the uterine arteries falls with advancing gestation, a finding attributed to progressive trophoblastic invasion and transformation of the uterine spiral arteries into large vessels of low resistance [2]. Failure to transform has been described in preeclampsia and fetal growth restriction, resulting in the use of a uterine artery blood flow Doppler assessment to screen for these pregnancy problems [3]. A recent review of reviews for preeclampsia screening methods demonstrated that uterine artery Doppler assessment as a stand-alone test had the best predictive value for the prediction of early-onset preeclampsia when compared to other tests with a moderate predictive value, such as increased body mass index (BMI), placental growth factor (PLGF), and placental protein 13 (PP13). The analysis also showed that no single biomarker met the standards required for a clinical screening test, but that models, that combined markers, were more promising for the prediction of preeclampsia [4]. In a recent randomized controlled trial, the use of such multimodal screening to determine the risk of preterm preeclampsia, followed by the prescription of low-dose Aspirin prophylaxis before 16 weeks’ gestation to the high-risk group, has been shown to halve the risk of preterm preeclampsia [5,6].

## 3. Uterine Artery Doppler Indices and Trophoblast Biology

The process of implantation, trophoblast development, and spiral artery transformation must involve many cellular and tissue processes to their effect. In view of the strong association between high uterine artery Doppler indices and the subsequent development of preeclampsia and fetal growth restriction, numerous authors have investigated trophoblast biology in samples obtained from pregnancies demonstrating high or low uterine artery Doppler resistance. Persistence of high resistance in the uterine artery Doppler indices in early pregnancy suggests that impaired trophoblast invasion and inadequate spiral artery remodeling has occurred [7].

### 3.1. Cell Injury and Apoptosis

Several studies have shown that placental tissues obtained from women with high-resistance uterine artery Doppler indices were more sensitive to apoptotic stimuli than placental tissue from women with normal indices [8,9,10]. Charolidi et al. examined the effect of tumor necrosis factor alpha (TNFα) on placental endothelial cell (PEC) apoptosis in the context of high vs. normal uterine artery Dopplers (Figure 1). They demonstrated that placental endothelial cells (PECs) from the high resistance index (RI) group exposed to TNFα had a 40% reduction in half-life compared to those from the normal RI group which were exposed to TNFα [10].

### 3.2. Cell Motility and Penetration

During early placentation, trophoblast cells invade the endometrium and differentiate to form the villous structure of the placenta. Extravillous trophoblast (EVT) migrates from said villi to attach the placenta to the decidual stroma cells (DSC) of the uterus. Interstitial EVT invades through the decidua into the myometrium, while endovascular EVT migrates into the lumen of the spiral arteries, replacing vascular smooth muscle cells, leading to spiral artery transformation [11]. James-Allan et al., assessed DSC function in pregnancies with high or normal uterine artery Doppler resistance indices in the first trimester. Their results showed that the chemoattraction of trophoblast cells by the DSC was dysfunctional when the DSC was gathered from a pregnancy with high uterine artery resistance, thus suggesting that there may be an interplay between the DSC and EVT in early pregnancy which might play a role in impaired trophoblast invasion related to high-resistance uterine artery blood flow [12].

### 3.3. Cell-Cell Interaction

Decidual natural killer (dNK) cells make up about 70% of the leukocytes found in the decidua in the first trimester [12]. dNK cells secrete a number of factors that disrupt vascular cell interactions and allow for vascular smooth muscle cells to migrate out of the spiral arteries [13,14,15]. dNK cells have also been shown to increase EVT motility by hepatocyte growth-factor secretion, leading to chemoattraction of the EVT to the sites of remodeling [16]. Wallace et al. demonstrated that dNK cells isolated during the first trimester from pregnancies with high uterine artery Dopplers have decreased ability to chemoattract trophoblast cells and induce the outgrowth of the EVT from the villi when compared to those from pregnancies with normal uterine artery resistance [17].

### 3.4. Oxidative Stress and Hypoxia

During the first trimester of pregnancy, the placenta develops in an environment with a relatively low oxygen supply of only 20mmHg at 8 weeks, rising to >50 mmHg at 12 weeks, as the maternal uterine artery blood flow increases [18]. Hypoxia-inducible nuclear factors HIF1α and HIF2α are master regulators of the hypoxia response in tissues, and are also expressed in the early-pregnancy placenta [19,20]. Levels of HIF1α are significantly lower in placentas gathered from pregnancies with high uterine artery resistance despite the expectation that placental hypoxia or oxidative stress would be associated with higher levels of HIF1α [8]. The latter tissues expressed an altered balance of antioxidant enzyme activity (lower glutathione peroxidase and higher superoxide dismutase activity) when compared with normal placental tissue. These findings all suggest that hypoxia and oxidative stress appear to be a physiological state in early pregnancy.

### 3.5. Altered Gene Expression

Two studies of first-trimester placental samples at the time of chorionic villous sampling demonstrated differences in gene expression when the woman subsequently developed preeclampsia or fetal growth restriction compared to those who had a normal pregnancy outcome [21,22]. One study which examined gene expression by microarray in first-trimester placentas demonstrated 26 genes were variably expressed in women with high-resistance uterine artery indices. The genes that were significantly differentially expressed included those mainly responsible for cell death/apoptosis, stress response, inflammatory/immune response, and the metabolic and cyclooxygenase pathways [8].

These findings all suggest a close and likely causative association between early-pregnancy uterine artery Doppler indices, trophoblast invasion, and the subsequent development of preeclampsia (Figure 2). High uterine artery resistance indices are predictive of the development of preeclampsia, and also influence trophoblast cell migration, apoptosis, motility, invasion, cell–cell interaction, response to oxidative stress, and gene expression.

## 4. Uterine Artery Blood Flow and Trophoblast Development—A Causality Paradox

The relationships demonstrated in the previous section between uterine artery blood flow and trophoblast cell function/behavior have been conventionally interpreted as reflecting the process of trophoblast invasion, causing decreased resistance to flow in the uterine artery. This hypothesis is propagated by the long-held belief that impaired trophoblast development results in poor spiral artery transformation and creates a predisposition to the development of preeclampsia. However, a number of recent findings have led to the re-evaluation of the cause–effect inference between trophoblast development and spiral artery transformation.

### 4.1. Abdominal Pregnancy

A recent case report described an advanced abdominal pregnancy where the placenta was implanted outside the uterus in the right-lateral pelvic side wall [24]. The authors took the initiative to describe the uterine artery Doppler findings, which demonstrated a low-resistance waveform consistent with a normal pregnancy placentation. The question this observation raises is how complete spiral artery transformation occurred in order to create the observed low-resistance uterine artery waveform consistent with a normal pregnancy when the placenta was implanted in an extra-uterine site. Case reports are often disregarded by scientists and clinical academics; however, in certain unique situations, the exception observed in a single clinical case may prove (or disprove) the rule. However, the latter finding has previously been consistently reported in extra-uterine pregnancy [25], and suggests that changes observed universally in the maternal uterine and spiral arteries that are reflected by uterine Doppler indices do not occur as a direct consequence of trophoblast invasion.

### 4.2. Late Pregnancy Uterine Artery Resistance Changes

Binder et al. studied 5887 pregnancies with longitudinal uterine Doppler assessment into the third trimester. They found that one-third of patients demonstrated a de-novo increase in uterine artery resistance in the late third trimester, having previously exhibited normal indices—and that this group had a 30% higher prevalence of preeclampsia [26]. If the conventional paradigm that normal trophoblast invasion causes spiral artery transformation and a decrease in uterine artery resistance is true, then the demonstration of an increase in third-trimester uterine artery resistance would require hypothetical “de-transformation” of the spiral arteries—an implausible biological phenomenon. An alternative explanation for the uterine artery Doppler findings in abdominal or late pregnancy is that observed variations are not caused by localized trophoblast invasion, but may, in fact, reflect maternal systemic vascular resistance changes [27,28].

### 4.3. Ophthalmic and Radial Artery Doppler Assessment

If, indeed, uterine artery waveform changes reflect maternal systemic hemodynamic perturbations rather than trophoblast invasion, then the Doppler assessment of non-uterine arterial vessels should mirror the findings in the uterine artery. Recent systematic reviews of Doppler assessment of the radial and ophthalmic arteries in pregnancy have demonstrated that these vessels also reduce their resistance with advancing gestation and demonstrate persistent high resistance in the first trimester in pregnancies at increased risk of preeclampsia [29,30]. These vessels seem to have the same ability to predict preeclampsia as the use of the uterine artery Doppler in isolation.

### 4.4. Pre-Pregnancy Maternal Systemic Vascular Resistance

Above all, the findings suggest that maternal systemic and uterine vascular resistance in pregnancy changes independently of the direct physical consequences of placental invasion and trophoblast cell behavior. If this were the case, it would imply that the biological associations described previously are inversely causal—that is to say, that increased uterine resistance and poor placental perfusion may result in impaired trophoblast invasion and function, rather than the other way around. The hypothesis that maternal systemic and uterine vascular impairment predates placental maldevelopment is supported by a recent prospective study assessing pre-pregnancy cardiovascular function in 530 women [31]. Women who subsequently developed preeclampsia had lower cardiac output and higher systemic vascular resistance in the pre-pregnancy state prior to the development of the trophoblast. The authors concluded that an altered pre-pregnancy hemodynamic phenotype was associated with the subsequent development of preeclampsia and/or fetal growth restriction.

## 5. Reviewing Evidence Supporting Placental Origins of Preeclampsia

Etiology and pathophysiology are two specific terms referring to distinct processes. The former refers to the origin of the disease, whilst pathophysiology refers to the biological mechanism by which the disease manifests clinical signs and symptoms. Whilst the role of the placenta in the pathophysiology of preeclampsia is indisputable, the observations supporting the hypothesis that abnormal placentation is the etiology of preeclampsia are restricted to spiral artery transformation (discussed above), abnormal placental histology, and fetal size.

### 5.1. Placental Histology in Preeclampsia

Preeclampsia has been attributed to maternal vascular malperfusion of the placental bed, characterized by myometrial/decidual vascular lesions (incomplete or absent remodeling of maternal spiral arteries) and, more commonly, placental villous lesions, such as accelerated villous maturation, distal villous hypoplasia, increased syncytial knots, and villous infarction [32,33]. Notably, these vascular and villous lesions are not specific to preeclampsia, and are also found in many other pregnancy disorders, such as fetal growth restriction, spontaneous preterm labor, placental abruption, and stillbirth [34,35]. A recent systematic review assessed the prevalence of vascular and villous lesions in preeclamptic and normal pregnancies [36]. The authors demonstrated that placental villous and vascular lesions were not seen in the majority of preeclamptic pregnancies (pooled prevalence of 45.2% and 38.2% in all studies, respectively) and were also seen in 10–20% of normal pregnancies. Interestingly, the authors also reported a three-fold overreporting of placental lesions in preeclampsia when the pathologist was unblinded to the pregnancy diagnosis, compared to blind reporting [37]. These findings show that the placental histological vascular and villous lesions previously presumed to be characteristic of preeclampsia are neither specific nor sensitive markers of the disorder.

### 5.2. Fetal Size in Preeclampsia

Fetal growth restriction is considered a typical feature of preeclampsia resulting from the primary placental dysfunction that causes the disorder. It is fetal growth restriction and associated hypoxemia that predispose to increased risk of fetal, neonatal, and long-term adverse outcomes [38,39,40]. Epidemiological studies demonstrate that the majority of preterm preeclampsia cases result in fetal growth restriction. However, over 80% of preeclampsia occurs at a term where the rate of large-for-gestational-age births are as common as small-for-gestational-age births (both 15%), and most neonates are of normal size, even after the exclusion of diabetic pregnancies [41,42]. The finding of normal or excessive fetal growth in the majority of term preeclampsia cases is not consistent with impaired trophoblast invasion with placental dysfunction being the primary etiological process in preeclampsia.

## 6. Evidence for Cardiovascular Origins of Preeclampsia

Earlier, we outlined the strong associations between uterine artery blood flow and trophoblast cell biology. The data also suggested that trophoblast invasion was not directly responsible for early pregnancy changes in uterine, ophthalmic, and radial artery hemodynamics, leaving us to entertain the alternative possibility that maternal cardiovascular function might be involved. If maternal cardiovascular function plays such an etiological role, this should also be evident from the epidemiology of preeclampsia.

### 6.1. Predisposing Factors for Preeclampsia

It is not entirely apparent how maternal clinical risk factors for preeclampsia, such as advanced maternal age, ethnicity, obesity, diabetes, hyperlipidemia, renal dysfunction, and chronic hypertension influence trophoblast invasion and spiral artery transformation. However, these are well-recognized risk factors for cardiovascular morbidity, with established and plausible biological mechanisms explaining their pathophysiological roles [43,44]. Apart from these conventional clinical risk factors, systematic reviews of genetic risk factors demonstrated that plasminogen activator inhibitor-1 (PAI-1) and FMS-related tyrosine kinase 1 (FLT1)—known to be linked with risks of coronary heart disease and heart failure—were also strongly associated with preeclampsia [45,46,47].

### 6.2. Pre-Pregnancy Cardiovascular Function in Preeclampsia

If maternal risk factors for preeclampsia and adult cardiovascular disease indeed work through the same mechanisms, then there should be evidence for pre-pregnancy cardiovascular impairment in women destined to develop preeclampsia. Although there is a paucity of such pre-conceptual studies, a recent study by Foo et al. longitudinally assessed cardiovascular function in 356 spontaneously conceived pregnancies in apparently healthy women, starting from preconception. The authors noted that the 15 (4.2%) women who developed preeclampsia and fetal growth restriction had lower cardiac output and higher total peripheral resistance before the pre-conceptual period compared to those with uneventful pregnancies [31]. These findings support the concept that suboptimal pre-pregnancy cardiovascular function may predispose the woman to impaired uterine artery blood flow and poor trophoblast development as precursors to the development of preeclampsia. This assertion is supported by pre-conceptual echocardiographic evaluation of formerly preeclamptic women assessing their risk of recurrent preeclampsia. Those with recurrent preeclampsia had lower cardiac left ventricular mass and cardiac stroke volume in the pre-pregnancy period compared to women who had a normal second pregnancy [48].

### 6.3. Abnormal Cardiovascular Function in Pregnancy and Preeclampsia

First-trimester maternal cardiovascular parameters, such as mean arterial blood pressure and uterine artery resistance, are key biomarkers in the ASPRE (Combined Multimarker Screening and Randomized Patient Treatment with Aspirin for Evidence-Based Preeclampsia Prevention) screening algorithm that have very high accuracy for the prediction of preeclampsia, especially of preterm onset [5]. More sophisticated echocardiographic assessment has shown cardiac remodeling, impaired hemodynamics, and diastolic dysfunction, both in the first trimester and at mid-pregnancy in women destined to develop preeclampsia compared to those with normal outcomes [49,50,51]. Even in uncomplicated pregnancies, there are well-documented hemodynamic changes which peak in the middle of the third trimester before the cardiac output falls and systemic vascular resistance increases—a paradoxical finding, in view of the fact that maternal respiratory and metabolic demands continue to increase with advancing gestation [52,53]. Echocardiographic studies have demonstrated that in apparently healthy women with normal pregnancies, there are signs of mild cardiac maladaptation to the volume overload, such as an excessive increase in the left ventricular remodeling with associated diastolic dysfunction in a small but significant proportion of cases at term [54,55]. It is apparent that even a normal pregnancy confers a previously unrealized significant workload on the maternal cardiovascular system, and that in some cases, this results in asymptomatic cardiac dysfunction (Figure 3).

### 6.4. Cardiovascular Function in Preeclampsia

At the clinical onset of preeclampsia, significant hemodynamic impairment, such as lower cardiac output, abnormal ventricular geometry, and diastolic dysfunction have been demonstrated by several maternal echocardiography studies [49,56,57,58]. Severe preterm disease is associated with a worse cardiovascular profile, which is in turn associated with higher rates of serious peripartum complications, such as pulmonary edema [59,60,61]. In keeping with these findings, a number of cardiovascular biomarkers, such as ANP-related proteins and Corin, have been shown to be altered in pregnancy in women with preeclampsia [61,62,63].

### 6.5. Abnormal Cardiovascular Function Persists after Preeclampsia

Birth is considered to be a “cure” for preeclampsia. While it is not in doubt that the signs, symptoms, and risks of preeclampsia regress in the vast majority of women within days after birth, the paradigm that delivery normalizes maternal health after preeclampsia is not supported by postpartum studies. The risk for developing chronic hypertension postnatally is much higher after preeclampsia than after normotensive pregnancy, with rates of up to 30% for chronic hypertension being reported at one year postpartum [64]. A large register-based study of over a million women confirmed a 20-fold increase in the rate of antihypertensive medication in pregnancies complicated by hypertensive disorders used within the first year after birth [65]. Even in women who are normotensive postpartum, asymptomatic moderate-severe cardiac dysfunction was significantly higher in preterm preeclampsia (56%) compared with term preeclampsia (14%) versus matched controls [66]. The issue of whether the latter findings were caused by preeclampsia or pre-existing was evaluated in a large Norwegian epidemiological study, which suggested that the increased postpartum cardiovascular risk after preeclampsia may most probably be due to pre-existing risk factors, rather than a detrimental effect of preeclampsia on the maternal cardiovascular system [67].

## 7. Analogy between Preeclampsia and Diabetes in Pregnancy

There are multiple clinical similarities between hypertension and diabetes in pregnancy, despite the fact that one is considered purely of placental origin and the other related to maternal pancreatic dysfunction [68,69,70]. Both disorders are diagnosed because of new-onset hypertension or hyperglycemia in pregnancy, predisposing risk factors are similar to adult-onset disease, the definitive treatment is birth, and both disorders leave a post-partum legacy of disease (Figure 4). The most convincing alignment between these two disorders is apparent when one considers the phenotypes of the disease. Pre-gestational diabetes and preterm pre-eclampsia both reflect primary organ dysfunction, and as such, are predisposed to pre-pregnancy disease and have a more severe, early-onset phenotype [71,72]. Gestational diabetes and term preeclampsia reflect normal organ function, being overcome by increased vascular/glycemia load late in pregnancy, and as such, are difficult to screen for and present themselves later in pregnancy with a milder phenotype [73].

## 8. Apparent Inconsistencies with Cardiovascular Origin Hypothesis

There are many iconic hypotheses applied to the placental origins of preeclampsia that stem out of clinical or epidemiological associations, such as those involving parity, change in partner, assisted reproductive technology, oocyte donation, and the “protective” effect of smoking. These hypotheses have assumed trophoblast origins of preeclampsia in their development, and a re-examination of their biological plausibility is justified.

### 8.1. Nulliparity

The risk of preeclampsia is about two times lower in multiparous women, and this has always been attributed to desensitization after exposure to paternal antigens in the placenta during previous pregnancies. Most epidemiological studies that report on parity and prevalence of preeclampsia do not account for the fact that, on average, multiparous women deliver approximately one week earlier than nulliparous women [74]. As shown in a recent randomized trial of induction of labor at 39 weeks’ gestation versus expectant management, the effect of this temporal difference is to reduce the prevalence of preeclampsia by about 40%, thereby accounting for a significant proportion of the different rates of preeclampsia with parity [75]. Cardiac assessment of pregnancy has also consistently demonstrated that parous women have a more favorable cardiovascular profile throughout pregnancy compared to nulliparous women [76,77]. Such cardiac programming is a well-accepted phenomenon in non-pregnancy physiology, and provides a biologically plausible rationale for different rates of preeclampsia with parity.

### 8.2. Change in Partner

Partner change is also considered by many to be a risk factor for preeclampsia, and is attributed to a maternal immune reaction against new paternal antigens expressed in the placenta [78]. Despite original studies indicating the importance of partner change as a risk factor, larger and more recent epidemiological studies in normal and assisted conception pregnancies have demonstrated that partner change is a proxy for lengthening inter-pregnancy intervals and advanced maternal age. Skjaerven and colleagues have demonstrated that a change of partner is not associated with an increased risk of preeclampsia after adjustment for the interval between births and maternal age [79,80].

### 8.3. Oocyte Donation

The increased risk of preeclampsia with assisted conceptions was shown to be attributable to advanced maternal age and oocyte donation [81,82]. The influence of oocyte donation as a risk factor is in alignment with disrupted immunological tolerance, as previously postulated, until one considers that women who conceive by egg donation tend to be older, affected by premature ovarian failure, or have mosaic Turner syndrome—and all these factors are known to increase cardiovascular risk [83,84].

### 8.4. Smoking

While it is well-established that smoking increases risk for poor fetal growth, its relation to preeclampsia is more controversial and not consistent with the placental origins hypothesis considering that a meta-analysis showed that smoking is inversely associated with the incidence of preeclampsia [85]. However, smoking is known to increase maternal carbon monoxide levels, which inhibit levels of sFlt-1 and increase those of PlGF—the opposite of what occurs in preeclampsia [86,87]. Furthermore, carbon monoxide also has a protracted hypotensive effect, which would decrease the diagnosis of high blood pressure and, as a consequence, paradoxically reduce the prevalence of preeclampsia whilst increasing the risk of poor fetal growth [88].

## 9. Conclusions

Placental cellular function and development may be controlled by maternal systemic and local uterine cardiovascular perfusion, rather than vice versa. The key role of the placenta and its discarded products in causing maternal endothelial disfunction during pregnancy is doubtless; nevertheless, the predisposition of women with cardiovascular dysfunction for developing preeclampsia, the development of cardiovascular dysfunction prior to disease onset, the predominance of cardiovascular signs/biology at presentation, and the long-term cardiovascular health risks post-partum all support the assertion that preeclampsia could be a primary cardiovascular disorder. Significant advances in screening, diagnosis, management, and post-partum cardiovascular health after preeclampsia may occur as we acknowledge this paradigm shift in disease causality.

## Figures and Tables

**Figure 1 ijms-20-03263-f001:**
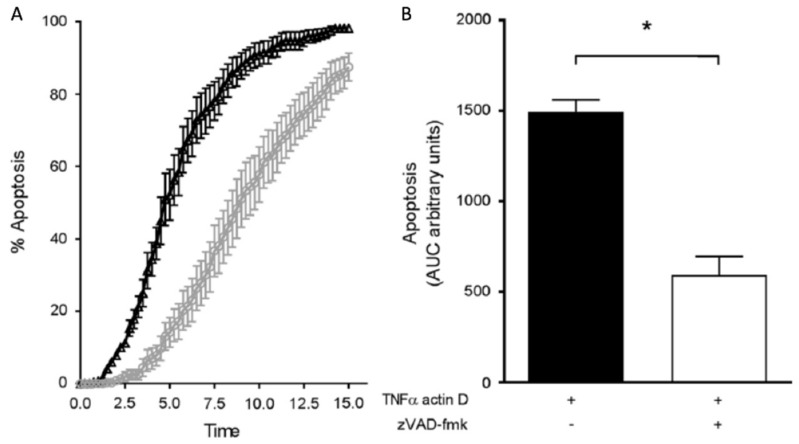
Apoptosis of first-trimester placental endothelial cells (PEC) from normal (normal RI) and high-resistance (high-RI) pregnancies in response to stimulation with TNFα and actinomycin D. First-trimester PEC were cultured with 30 ng/mL TNFα and 800 ng/mL actinomycin D. Images were taken every 15 min over 15 h. (**a**) The kinetics of the induction of apoptosis for PEC, high RI (n = 8 mean ± SEM, black symbols) and normal RI (n = 8 mean ± SEM, grey symbols). (**b**) In a separate cohort, normal-RI PEC were incubated with TNFα and actinomycin D alone (n = 4), as well as in the presence of the broad-spectrum caspase inhibitor, zVAD-fmk (n = 4). The results are expressed as mean ± SEM and * p < 0.05. Adapted with permission from Charolidi et al. [9].

**Figure 2 ijms-20-03263-f002:**
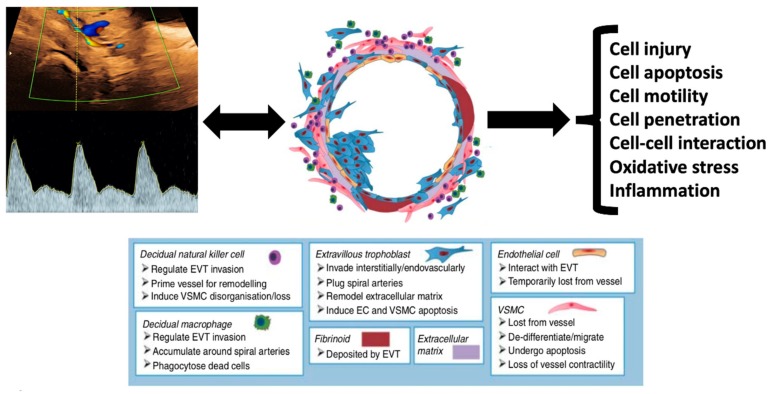
Relationship between uterine artery perfusion and cellular function. There are strong associations between maternal uterine artery perfusion and placental cellular function and behaviour. High-resistance Doppler indices (poor placental perfusion) is related to abnormalities of cell motility, penetration, and cell–cell interaction, as well as increased rates of oxidative stress, inflammation, cellular injury, and apoptosis. Adapted with permission from Cartwright et al. [23].

**Figure 3 ijms-20-03263-f003:**
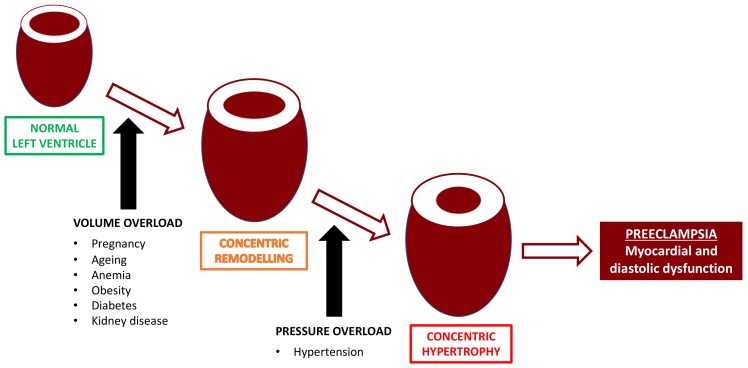
Left ventricle remodelling caused by pregnancy and preeclampsia. Alteration in (volume and pressure) loading conditions and the interaction with mechanical and neurohormonal factors results in ventricular remodeling. At an organ level, remodeling refers to changes in ventricular geometry, volume, and mass. Although remodelling is compensatory in certain pressure and volume overload conditions, progressive ventricular remodeling is ultimately a maladaptive process, contributing to the progression of symptomatic heart failure and an adverse outcome.

**Figure 4 ijms-20-03263-f004:**
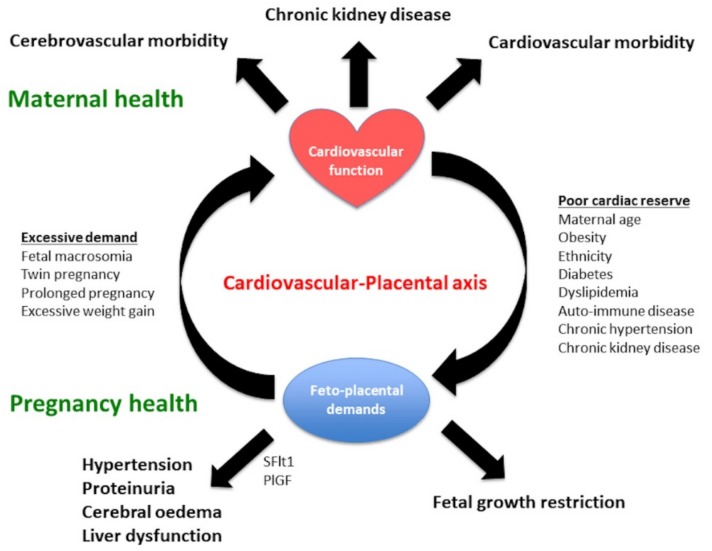
Interaction between maternal cardiovascular function and placental function, maternal health, and fetal well-being. Placental oxidative stress or hypoxia is related to the relative balance of cardiovascular functional reserve and the cardiovascular volume/resistance load of pregnancy. The final common pathway that results in the signs and symptoms of preeclampsia involves the release of placental vasoactive substances. Adapted with permission from Thilaganathan and Kalafat. Hypertension. 2019;73:522–531 [68].

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
