# Peer review of "Preeclampsia: The Relationship between Uterine Artery Blood Flow and Trophoblast Function"

_ijms, 2019, doi:10.3390/ijms20133263_

Reviewer 1 Report

There can be no doubt that preeclampsia is a complex syndrome involving poor trophoblast invasion, placentation and a maternal response that drives the appearance of the physiological symptoms. This review argues strongly that maternal cardiovascular changes underpin the development of preeclampsia rather that placental development. It offers an alternative hypothesis and summarises recent findings to support this. It should be published but in my opinion needs to be moderated, specifically the conclusions in section 9, which need to reflect the role of the placenta in preeclampsia.

There is a large volume of evidence missing from this review, and that is the role of placental debris and the effect that this has on maternal endothelium. Multinuclear aggregates, microvesicles, membrane fragments, exosomes are all shed from the placenta and there is good evidence that this is altered in preeclampsia and drives the endothelial changes underpinning the syndrome.

Additionally, little is made of immunological basis of preeclampsia. There is considerable evidence of altered cytokine responses and a pro-inflammatory state but this seems to largely overlooked. The review has a focus at the physiological level but at a biochemical level there is evidence that molecular changes in the trophoblast leads to release of potent mediators that correlate with the progression and severity of the disease. sFLT and PlGF are mentioned but there are others that argue strongly of placental involvement.

In my opinion, our understanding of preeclampsia was held back for many years as the dogma was that it was a “ placental disease”. More recently, the development of a model that describes a placental stimulus combined with a maternal reaction seems to fit with most of the evidence. In my view this review swings to far the other way and whilst is adds value by promoting discussion I think it too heavily states its’ claim given the evidence presented.  

Author Response

Point 1:  There can be no doubt that preeclampsia is a complex syndrome involving poor trophoblast invasion, placentation and a maternal response that drives the appearance of the physiological symptoms. This review argues strongly that maternal cardiovascular changes underpin the development of preeclampsia rather that placental development. It offers an alternative hypothesis and summarises recent findings to support this. It should be published but in my opinion needs to be moderated, specifically the conclusions in section 9, which need to reflect the role of the placenta in preeclampsia.

There is a large volume of evidence missing from this review, and that is the role of placental debris and the effect that this has on maternal endothelium. Multinuclear aggregates, microvesicles, membrane fragments, exosomes are all shed from the placenta and there is good evidence that this is altered in preeclampsia and drives the endothelial changes underpinning the syndrome.

Additionally, little is made of immunological basis of preeclampsia. There is considerable evidence of altered cytokine responses and a pro-inflammatory state but this seems to largely overlooked. The review has a focus at the physiological level but at a biochemical level there is evidence that molecular changes in the trophoblast leads to release of potent mediators that correlate with the progression and severity of the disease. sFLT and PlGF are mentioned but there are others that argue strongly of placental involvement.

In my opinion, our understanding of preeclampsia was held back for many years as the dogma was that it was a “ placental disease”. More recently, the development of a model that describes a placental stimulus combined with a maternal reaction seems to fit with most of the evidence. In my view this review swings to far the other way and whilst is adds value by promoting discussion I think it too heavily states its’ claim given the evidence presented.  

Point 1: Thank you for your constructive comment. We mitigated the conclusions in section 9 following your indications. We agree that preeclampsia is a complex disease where different factors occur and interact with one another. Among those, placenta is still one of the most significant element in causing the disease during pregnancy, but it is not enough to explain many other factors related to preeclampsia and maternal health. Without minimizing other hypotheses, our review would like to focus on an alternative view that has been supported by growing evidence, but it should still be fully validated. 

Reviewer 2 Report

1) Line 30-32. “Spiral arterioles … fetoplacental unit.” Reference missing.

2) Line 51-52. “…, but that models that combined markers were more promising for the prediction of preeclampsia.” Should be “… that models with combined markers…”

3) Line 99-101. “The latter tissues expressed an altered balance of … when compared with normal placental tissue.” Reference missing.

4) Line 165-167. “Women who subsequently developed… development of the trophoblast.” Reference missing.

5) Line 185-187. “The authors demonstrated that placental villous and vascular lesions were not seen in the majority (~60%) of preeclamptic pregnancies and were also seen in 10-20% of normal pregnancies.” this sentence doesn’t make sense.

6) Line 280-284. “Pre-gestational diabetes and preterm pre-eclampsia both reflect … present late in pregnancy with a milder phenotype.” Reference missing.

7) Line 294-296. “Most epidemiological … on average multiparous women deliver approximately one week earlier than nulliparous women.” Reference missing.

8) Line 305-306. “Partner change is … against new paternal antigens expressed in the placenta.” Reference missing.

Author Response

1)   Line 30-32. “Spiral arterioles … fetoplacental unit.” Reference missing.

Point 1: Reference inserted!

2)   Line 51-52. “…, but that modelsthatcombined markers were more promising for the prediction of preeclampsia.” Should be “… that models withcombined markers…”

Point 2: In this case it is “models, that combined markers”

3)   Line 99-101. “The latter tissues expressed an altered balance of … when compared with normal placental tissue.” Reference missing.

Point 3: The sentence is referencing the previous sentence with reference number [8]

4)   Line 165-167. “Women who subsequently developed… development of the trophoblast.” Reference missing.

Point 4: Here we are referring to the patients from the study mentioned in the previous sentence with reference [30]

5)   Line 185-187. “The authors demonstrated that placental villous and vascular lesions were not seenin the majority (~60%) of preeclamptic pregnancies and were also seen in 10-20% of normal pregnancies.” this sentence doesn’t make sense.

Point 5: Thank you for this revision. We rewrite the sentence giving more specific information in order to make it clearer.

6)    Line 280-284. “Pre-gestational diabetes and preterm pre-eclampsia both reflect … present late in pregnancy with a milder phenotype.” Reference missing.

Point 6: Thank you for your suggestion. We added the following references:

 Weissgerber TL, Mudd LM. Preeclampsia and diabetes. Curr Diab Rep. 2015 Mar;15(3):9. 

Alexopoulos AS, Blair R, Peters AL. Management of Preexisting Diabetes in Pregnancy: A Review. JAMA. 2019 May 14;321(18):1811-1819. 

Nzelu D, Dumitrascu-Biris D, Nicolaides KH, Kametas NA. Chronic hypertension:  first-trimester blood pressure control and likelihood of severe hypertension, preeclampsia, and small for gestational age. Am J Obstet Gynecol. 2018 Mar;218(3):337.e1-337.e7

7)    Line 294-296. “Most epidemiological … on average multiparous women deliver approximately one week earlier than nulliparous women.” Reference missing.

Point 7:  Thank you for your correction. We provided a new reference “Smith GC. Use of time to event analysis to estimate the normal duration of human pregnancy. Hum Reprod. 2001 Jul;16(7):1497-500”. 

8)    Line 305-306. “Partner change is … against new paternal antigens expressed in the placenta.” Reference missing.

Point 8: Done. The added  reference was  “Li DK, Wi S. Changing paternity and the risk of preeclampsia/eclampsia in the subsequent pregnancy.  American journal of epidemiology. 2000;151(1):57–62”. 
